# Comparison of Hemodynamic and Cerebral Oxygenation Responses during Exercise between Normal-Weight and Overweight Men

**DOI:** 10.3390/healthcare11060923

**Published:** 2023-03-22

**Authors:** Szu-Hui Wang, Hui-Ling Lin, Chung-Chi Huang, Yen-Huey Chen

**Affiliations:** 1Department of Respiratory Therapy, Chiayi Chang Gung Memorial Hospital, Chiayi 61363, Taiwan; 2Department of Respiratory Therapy, College of Medicine, Chang Gung University, Taoyuan 33301, Taiwan; 3Department of Respiratory Therapy, Chang Gung University of Science and Technology, Chiayi 61363, Taiwan; 4Division of Pulmonary and Critical Care Medicine, Chang Gung Memorial Hospital, Linkou. 5, Fu-Hsin St. Gweishan, Taoyuan 33353, Taiwan

**Keywords:** obesity, exercise, cerebral oxygenation, hemodynamics

## Abstract

Obesity has negative impacts on cardiovascular function and may increase cerebrovascular complications during exercise. We compared hemodynamic and cerebral oxygen changes during high-intensity exercise between overweight (OW) and normal-weight (NW) individuals. Eighteen NW and fourteen OW male individuals performed high-intensity (70% of peak oxygen uptake, VO_2_peak) cycling exercises for 30 min. Hemodynamics were measured using a bioelectrical impedance device, and cerebral oxygenation status was measured using a near-infrared spectrophotometer during and after exercise. The VO_2_peak of NW individuals was significantly higher than that of OW individuals (41.3 ± 5.7 vs. 30.0 ± 5.0 mL/min/kg, respectively; *p* < 0.05). During the 30 min exercise, both groups exhibited an increase in oxygenated hemoglobin (O_2_Hb) (*p* < 0.001), deoxygenated hemoglobin (*p* < 0.001), and cardiac output with increasing time. Post-exercise, cardiac output and systemic vascular resistance were significantly higher in the OW group than in the NW group (*p* < 0.05). The O_2_Hb in the NW group was significantly higher at post-exercise times of 20 min (13.9 ± 7.0 μmol/L) and 30 min (12.3 ± 8.7 μmol/L) than that in the OW group (1.0 ± 13.1 μmol/L and 0.6 ± 10.0 μmol/L, respectively; *p* = 0.024 vs. 0.023, respectively). OW participants demonstrated lower cerebral oxygenation and higher vascular resistance in the post-exercise phase than non-OW subjects. These physiological responses should be considered while engaging OW and obese individuals in vigorous exercise.

## 1. Introduction

Currently, obesity is a global problem, with 60% of adults in the US and Europe being overweight or obese [1]. According to the World Health Organization (WHO), a body mass index (BMI) of over 25 kg/m^2^ is defined as overweight, and individuals with a BMI of over 30 kg/m^2^ are considered obese [2]. However, the WHO definition is based primarily on criteria derived from studies involving populations of European origin. It has been suggested that the BMI cut-off point (≥30 kg/m^2^) might be too high for Asians, thereby underestimating the associated health risks [2,3]. In some Asian countries, obesity is defined as a BMI ≥ 27 kg/m^2^, overweight as a BMI of 24–27 kg/m^2^, and a normal BMI as 18.5 to <24 kg/m^2^ [4]. 

Obesity is an important risk factor for many diseases, including metabolic syndrome, hypertension, diabetes, cardiovascular disease, and cancer [4,5]. It leads to excessive fat accumulation in blood vessels, resulting in atherosclerosis, along with an increase in blood flow resistance and cardiac afterload. This results in left ventricular hypertrophy and cardiovascular dysfunction. Obesity also affects physical fitness in individuals who are not sick or asymptomatic [5]. Vella et al. compared exercise performance in normal-weight and obese participants and found that obese participants had lower left ventricular ejection fraction (LVEF) and peak oxygen consumption during exercise tests [6]. 

In humans, the brain represents only 2–3% of the total body mass, requires 15% of the cardiac output, and consumes 20% of the available O_2_ under normal conditions. The high metabolic rate of the brain, combined with limited energy stores, highlights the importance of cerebral blood flow for nutrient and O_2_ delivery, as well as for the removal of cellular, metabolic, or toxic by-products [7]. In addition to cardiovascular dysfunction, obesity also has profound effects on brain structure and vasculature owing to metabolic disturbances and blood flow dysregulation [8]. Studies have shown that obesity increases the stiffness of the carotid artery, which may affect blood supply to the brain at rest [9,10]. However, most of these studies were performed in the resting state. Thus, the mechanisms underlying the interaction between cerebral oxygen and the central hemodynamic system remain unclear. 

Cerebral oxygenation can serve as an index of cerebrovascular function, as it depends on blood flow and endothelial integrity [11]. Near-infrared spectroscopy (NIRS) assesses cerebral oxygen delivery and uptake by cerebral tissues through the measurement of oxy- (O_2_Hb) and deoxygenated hemoglobin (HHb), which provides information on cerebrovascular function [12]. During exercise, cerebral oxygenation increases in healthy humans due to an increase in regional total hemoglobin (tHb) and O2Hb [13]. This increase in oxygen delivery is important because brain function can deteriorate when oxygenation is reduced. Subjects with obesity are at risk of cerebrovascular dysfunction; however, few studies have examined their cerebral oxygenation response to exercise.

In order to improve cardiopulmonary function and physical fitness, the American College of Sports Medicine recommends that adults engage in at least 30 min of moderate-intensity exercise at least five times a week or at least 25 min of vigorous-intensity exercise at least three times a week [14]. However, owing to the negative effects of obesity on the cardiovascular system, obese people are at a higher risk of cardio- and cerebrovascular accidents when exercising than non-obese people, especially during moderate or vigorous exercise. In addition, cardiovascular function and cerebral oxygenation status are strongly related to recovery from exercise. A previous study reported that compared with a control group, obese individuals have higher sympathetic activity, as evaluated by an exercise recovery index following volitional exhaustion during an incremental exercise test [15]. Obesity-related slowing in recovery of oxygen uptake and cardiovascular response may serve as secondary measures of cardiovascular fitness and disease risk [16]. Although many studies have examined cardiovascular responses following exercise, the influence of submaximal work on subsequent recovery of cerebral oxygenation status has not been extensively evaluated.

The primary purpose of this study was to compare the cardiovascular and cerebral hemodynamic responses to submaximal work in young male participants with normal weight and overweight. The secondary objective was to investigate the role of overweight in recovery from submaximal work and the association between cerebral oxygenation and hemodynamic indices.

## 2. Materials and Methods

### 2.1. Participants

This was a cross-sectional, interventional, controlled study. Thirty-two healthy university students were recruited from a university campus between June and December 2016. Subjects were divided into two groups based on their BMI. Subjects with a BMI which was in the normal range (18.5 ≤ BMI < 24 kg/m^2^) were in the non-overweight (NW) group (*n* = 18), and those with a BMI ≥ 24 kg/m^2^ were in the overweight (OW) group (*n* = 14). The inclusion criteria for all participants were as follows: (1) age 20–40 years and (2) male sex. The exclusion criteria were as follows: (1) engaged in regular exercise during the past 6 months (i.e., moderate or vigorous exercise for ≥20–30 min/time, ≥3 times/week); (2) any cardiovascular, metabolic, neuromuscular, respiratory, renal, or other systemic diseases; and (3) any musculoskeletal or joint problems in the upper or lower extremities. All participants completed a self-reported physical activity questionnaire (IPAQ) [17], which was administered to exclude potential volunteers who engaged in regular physical activities. This study was approved by the Institutional Review Board of Chang Gung Hospital (201600418B0). This study was performed in accordance with the principles of the Declaration of Helsinki. Written informed consent was obtained from all of the participants prior to inclusion.

### 2.2. Procedures

All participants reported to the laboratory for two visits which were separated by at least 1 week. At the first visit, participants underwent anthropometric assessments and incremental exercise tests in that order. On the second visit, the subjects performed 30 min of cycling exercise. Participants were instructed to refrain from alcohol and caffeine for 12 h before each visit. Participants were also required to maintain normal hydration, to refrain from strenuous exercise for at least 24 h prior to each visit, and to report to the laboratory at least 3 h after their last meal. 

At the first visit, anthropometric assessments were performed using bioelectrical impedance analysis (Inbody 720, Inbody Co., Seoul, Republic of Korea). The subjects were required to stand barefoot on the metal sole plates of the device while holding electrodes, one in each hand, and a safe electrical signal (50 kHz, 800 µA) was sent through the body. The circumferences of the waist and hip were measured using an unstretchable metric tape. Waist circumference was measured by passing the measuring tape through the midpoint between the superior iliac crest and lowest rib. Hip circumference was measured at maximum protrusion. An incremental exercise test was performed using a cycle ergometer (VIAspritn 150P; Carefusion Corp., San Diego, CA, USA). An individualized protocol began with a 3 min rest to familiarize the participants with the equipment, followed by a 3 min warm-up at a workload of 20 W. Power was gradually increased at a rate of 20–30 W/min until the participant was exhausted. Cycling cadence was maintained at 60–80 rpm. Gas exchange was measured continuously during the exercise test session using a metabolic system (Vmax Encore Metabolic Care; Carefusion Corp., USA). The following data were recorded every 15 s: oxygen uptake (V˙O_2_ mL/min), product of carbon dioxide (VCO_2_), respiratory exchange ratio (RER), minute volume V˙E (L/min), and respiratory rate (breaths/min). Electrocardiography (ECG) was used to monitor the heart rate, utilizing a 12-lead system before and during the test for safety. The incremental exercise test was terminated when subjects met three of the following four criteria: (1) a perceived exertion score of ≥17 on the Borg scale (scale 6–20); (2) respiratory exchange ratio ≥1.1; (3) an increase in heart rate to >90% of age-predicted maximal heart rate (220−age); and (4) volitional exhaustion in accordance with the American Thoracic Society standard test termination criteria [18]. The anaerobic threshold (AT) was also defined by the following criteria: (1) steeper increase in VCO_2_ as compared to VO_2_; (2) a respiratory exchange ratio ≥0.95–1.0; (3) PETO_2_ increase; and (4) VE/VO_2_ increase [18]. In the second visit, the participants performed 30 min of continuous cycling exercise at 70% of the maximum workload, which was obtained from the incremental exercise test during the first visit. Exercise of this intensity and duration is recommended in order to reduce body weight in adults [19]. Briefly, the subjects began with a 5-min warm-up at a pedaling cadence of 60 rpm, after which the workload was increased to 70% of the maximum workload. The subjects were asked to exercise at this intensity for 30 min. After task termination, the subjects pedaled at a free load for a 5 min cool down period. The subjects then rested on a chair for 30 min during the recovery phase. During the incremental exercise test (first visit) and 30 min cycling exercise (second visit), central hemodynamic and cerebral oxygenation responses were continuously measured.

### 2.3. Measurement

#### 2.3.1. Central Hemodynamic Measurements

The central hemodynamic responses of the subjects were continuously monitored using the bioimpedance method (PhysioFlow, Manatec Biomedical, Paris, France), following the procedure described in a previous study [20]. This method has been validated during maximal incremental exercises [21] and has been used in obese subjects [6]. 

PhysioFlow values were synchronized and mediated every 10 s. The transthoracic bioimpedance system (PhysioFlow) measures the variation in impedance (Z) to high-frequency, low-amperage alternating electrical current using two electrodes in the thoracic (xiphoid process) and two electrodes in the neck area. The physiological principle is based on the change in impedance, which is related to systolic and diastolic fluid variations in the thorax [22]. The derivative of the waveform (dZ) is related to the contractility and systolic volume. The waveform was also related to the atrial and ventricular systole and diastolic function. Therefore, the Physioflow device provided the following hemodynamic parameters: heart rate (beats/min), stroke volume (ml), cardiac output (L/min), heart rate (HR), stroke volume (SV), stroke volume index (SVI), cardiac output (CO), cardiac index (CI), systemic vascular resistance (SVR), and systemic vascular resistance index (SVRI). SV was obtained using the following equation: SV = *k* · [(dZ/dtmax)/(Zmax − Zmin)] · W (thoracic flow inversion time_cal_), where *k* is a constant, W is a proprietary correction algorithm, and “cal” indicates that the value was obtained during auto-calibration [22]. Some parameters were calculated according to the following equation: CO (L/min) = SV · HR; CI (L/min/m^2^) = CO/body surface area; SVR (dynes∗sec/cm^5^) = 80 ∗ (mean blood pressure-central venous pressure)/CO (the central venous pressure was set to 7 mmHg); systemic vascular resistance index (SVRi) (dynes∗sec/cm^5^∗m^2^) = SVR/body surface area. Hemodynamic parameters were monitored at rest, during exercise, and during the recovery phase.

#### 2.3.2. Cerebral Oxygenation Measurements

Cerebral hemodynamic responses were obtained by measuring changes in the relative concentration (Δμmol) of oxygenated hemoglobin (O_2_Hb) and deoxygenated hemoglobin (HHb) using a continuous-wave NIRS system (PortaMon, Artinis, The Netherlands) during the rest, exercise, and recovery phases. NIRS technology emits near-infrared light into tissues, where it is either scattered within the tissue or absorbed by chromophores such as oxygenated hemoglobin (O_2_Hb) and deoxygenated hemoglobin (HHb). The levels of HHb and O_2_H within tissues can be assessed by measuring the returned light at specific wavelengths. O_2_Hb represents the status of tissue oxygenation and HHb represents the status of local tissue deoxygenation from O_2_ extraction. The total hemoglobin (tHb) is the sum of O_2_H and HHb and is an indicator of regional blood volume. A probe was placed on the left forehead to measure cerebral oxygenation. Elastic bandages were used to shield the probe from ambient light. The relative concentration changes (Δ) in resting baseline oxyhemoglobin (ΔO_2_Hb), deoxyhemoglobin (ΔHHb), and total hemoglobin (ΔtHb) were measured. Data were sampled at 10 Hz during the rest, exercise, and recovery periods [23].

### 2.4. Statistical Analysis

Sample size estimates were generated based on the detection of a 10% difference between NW and obese subjects for the stroke volume index [24] and submaximal cardiac index [25]. At least 15 subjects per group were required, assuming a statistical power of 0.8 and an α of 0.05. Considering the possibility of 10–20% of individuals dropping out, we set the sample size to 18 subjects. 

Statistical analyses were performed using the IBM SPSS for Windows (version 22.0; IBM Corp., Armonk, NY, USA). Normality was confirmed using the Shapiro–Wilk test. The results are expressed as the mean ± standard deviation for nominal distributions and the median (interquartile range) for nonparametric distributions. The differences in physical characteristics between the NW and OW groups were compared using Student’s *t*-test, and the 95% confidence intervals (CIs) for the comparisons were reported as outcomes. Within-group differences were evaluated using repeated-measures analysis of variance for the variables at different time points (rest, exercise, and post-exercise period). A two-way ANOVA with repeated measures (group × time) was performed on hemodynamic and cerebral oxygenation variables in order to compare group differences in response to exercise. When a significant group-by-time interaction was present, post hoc comparisons were conducted between groups at each time point (Bonferroni correction and independent *t*-test). The effect size (partial η^2^) from the repeated ANOVA test was reported as the outcome. The magnitude of this effect size was interpreted as follows: small (η^2^ = 0.01–0.08), medium (η^2^ = 0.09–0.24), and large effect (η^2^ = 0.25 and over). The relationships between VO_2_ peak and maximal cerebral oxygenation variables during the exercise test were assessed using the Pearson’s coefficient of correlation. All *p* values were two-sided, and significance was set at *p* < 0.05.

## 3. Results

### 3.1. Clinical and Anthropometric Characteristics

A total of 32 participants (18 in the NW group and 14 in the OW group) completed the study (Figure 1). The participants’ basic information is presented in Table 1. There was no significant difference in mean age between the NW and OW groups (21.6 ± 0.7 and 21.8 ± 1.0 years, respectively; *p* = 0.543). Regarding body composition, there was no significant difference in muscle mass between the NW and OW groups (31.1 ± 1.9 vs. 32.0 ± 4.4 kg, respectively; *p* = 0.475). The average body fat rate of the OW group was 30.0 ± 7.8; this was significantly higher than that of the NW group (19.3 ± 4.8%) (*p* < 0.001) (Table 1).

Table 2 lists the variables used in the incremental exercise test. There was no significant difference in resting heart rate between the two groups (76.0 ± 12.1 vs. 82.6 ± 9.8 bpm for the NW and OW groups, respectively; *p* = 0.174). The resting diastolic blood pressure in the OW group was significantly higher than that in the NW group (88.4 ± 9.1 vs. 76.2 ± 8.3 mmHg, *p* = 0.004). The anaerobic threshold load of the OW group (64.7 ± 20.1 watts) was significantly lower than that of the NW group (87.2 ± 26.8 watts, *p* = 0.032). The peak oxygen uptake (normalized to body weight) of the NW group was significantly greater than that of the OW group (41.3 ± 5.7 and 30.0 ± 5.0 mL/min/kg, respectively; *p* < 0.001). Furthermore, the peak oxygen uptake/predicted value (%) of the NW group was also significantly greater than that of the OW group (80.1 ± 14.3 and 66.0 ± 7.5%, respectively; *p* = 0.008). 

When pooling the data from the two groups, peak oxygen uptake was positively correlated with cerebral ΔO_2_Hb (r = 0.39, *p* = 0.044), ΔHHb (r = 0.469, *p* =0.014), and ΔtHb (r = 0.442, *p* =0.021) (Table 3).

### 3.2. Cardiovascular and Cerebral Hemodynamics Response during 30 min Continuous Exercise

The trends in cerebral hemodynamics during and after 30 min of continuous exercise are presented in Figure 2a,b, respectively. The main effect of time was significantly different for ΔO_2_Hb (F = 12.2, *p* < 0.001, η^2^ = 0.526) (Figure 2a) and ΔHHb (F = 4.0, *p* < 0.001, η^2^ = 0.267) (Figure 2b). In both the NW and OW groups, the ΔO_2_Hb significantly increased from resting to 30 min of cycling exercise (from 6.4 ± 6.4 to 27.6 ± 10.8 μmol/L in NW, *p* < 0.001; from 7.7 ± 0.45 to 17.6 ± 1.0 L/min in OW, *p* < 0.001). The main effects of group were significantly different for ΔO_2_Hb (F = 5.2, *p* = 0.044, η^2^ = 0.321) (Figure 2a) and ΔHHb (F = 5.1, *p* = 0.046, η^2^ = 0.315) (Figure 2b). At 30 min post-exercise, the ΔO_2_Hb in the NW group (13.9 ± 7.0 μmol/L) was significantly higher than that in the OW group (0.6 ± 10.0 μmol/L, respectively) (*p* = 0.049) (Figure 2a). Additionally, at 30 min post-exercise, the ΔHHb in the NW group (2.1 ± 2.8 μmol/L) was significantly higher than that in the OW group (−3.0 ± 5.4 μmol/L)(*p* = 0.046) (Figure 2b) (Appendix A). 

The cardiac output, stroke volume, and SVRi responses during and after continuous exercise are shown in Figure 3a–c. The main effect of time was significantly different in the measurement of cardiac output (F = 265, *p* < 0.001, η^2^ = 0.903), stroke volume (F = 92, *p* < 0.001, η^2^ = 0.807), and SVRi (F = 151, *p* < 0.001, η^2^ = 0.873). In both the NW and OW groups, the CO significantly increased from resting to 30 min of cycling exercise (from 5.4 ± 0.37 to 16.9 ± 0.85 L/min in NW, *p* < 0.001; from 7.7 ± 0.45 to 17.6 ± 1.0 L/min in OW, *p* < 0.001) (Figure 3a). The SVRi at 30 min of exercise was significantly lower than that in their resting status in the NW (from 2395 ± 485 to 752 ± 125 dynes ∗sec/cm^5^ ∗ m^2^, *p* = 0.001) and OW groups (from 2140 ± 604 to 922 ± 222 dynes ∗ sec/cm^5^ ∗ m^2^, *p* = 0.001) (Figure 3c). In the OW group, the resting CO (7.7 ± 0.45 vs. 5.4 ± 0.37L for the OW and NW groups, respectively; *p* = 0.009) and SV (84.8 ± 4.2 vs. 72.8 ± 3.6 mL/beat for the OW and NW groups, respectively; *p* = 0.000) were significantly higher compared with those in the NW group. The CO and SV during the exercise and post-exercise phases tended to be higher in the OW group than in the NW group. However, no significant differences were found between the NW and OW groups. Moreover, no significant difference was found between the groups in the measurement of SVRi during and after the cycling exercise (Appendix A).

## 4. Discussion

This study examined the effects of acute exercise on hemodynamic and cerebral oxygen status. Our study showed that OW individuals had lower cardiorespiratory fitness than non-OW individuals. In addition, overweight individuals experienced cerebral and hemodynamic dysfunction during and after a 30 min continuous exercise session. 

Cardiometabolic risk factors are associated with increased risks of cerebrovascular disease and cardiopulmonary dysfunction [4,5]. Participants in the NW group had significantly higher cardiorespiratory fitness than those in the OW group. Myung et al. reported that body fat percentage, but not skeletal muscle mass, was inversely correlated to VO_2_peak in obesity individuals [26]. In our study, although muscle mass was not significantly different between the NW and OW groups, the proportion of skeletal muscles to the total body weight was lower, whereas the body fat percentage was significantly higher in the OW group. Excessive body fat increases cardiovascular workload during exercise, resulting in a decrease in exercise capacity. We also observed that the waist circumference (WC), waist/hip ratio (WHR), and body fat percentage (BF) were significantly higher in the OW group than in the NW group. WC and WHR are indicators of abdominal body fat, which are associated with cardiometabolic disease and are predictors of mortality [4]. Adipose tissues in the abdomen have been reported to limit movement of the diaphragm, which may lead to decreased lung volume, early carbon dioxide retention, and hypoxia during exercise, thus resulting in exercise intolerance [4].

In our study, CO was significantly higher at rest in the OW group than in the NW group. In obese individuals, the increased metabolic demand imposed by excessive adipose tissue results in hyperdynamic circulation and increased blood volume [27]. This results in an increased stroke volume and cardiac output. The increased cardiac output in obese patients meets the metabolic demand for excess adipose tissue. Although the heart rate is higher in obese individuals owing to increased sympathetic activation, increased cardiac output is related to an increase in stroke volume [27,28]. We also observed elevated hemodynamic responses during and after exercise. However, no statistical difference in hemodynamic variables was found between the OW and NW groups. As mentioned above, the elevated hemodynamic status at rest in obese individuals may remain consistently high in order to meet the increasing oxygen demand during exercise. Zeiler et al. examined hemodynamic responses after exercise and reported that stroke volume and cardiac output were higher in the obese group than in the non-obese group at rest and throughout the 60 min post-exercise period [29]. Cavuoto et al. reported a higher cardiac output a submaximal exercise in obese individuals when compared to those in the non-obese group [30]. The discrepancies among studies might be explained by different factors, including the level of obesity, exercise intensity, and/or other factors. Collectively, these reports suggest that adaptations to cardiac output and stroke volume during exercise mirror those observed during rest in overweight adults. 

Hiura et al. reported a significant increase in cerebral ΔO2Hb and cerebral blood flow in healthy subjects during 15 min of low-intensity exercise [31]. In a review study, it was summarized that ΔO_2_Hb increases from low-, moderate-, and vigorous-intensity endurance exercise, and reaches a steady state in vigorous-intensity exercise. In our study, we also observed a significant increase in cerebral ΔO_2_Hb levels from rest to exercise for 30 min. The magnitude of the effect of time was 0.526, which is considered large. Exercise can be regarded as an ultimate integrative stimulus for the brain to regulate cerebral blood flow and oxygen uptake. During exercise, oxygen uptake, cardiac output, sympathetic nerve activity, and brain uptake increase constantly challenge the sufficient delivery of cerebral blood flow to meet metabolic and oxygen demands [7]. An increase in cardiac output in our study from baseline to 30 min of exercise suggests that increased cerebral oxygenation may be related to an increase in systemic circulation during exercise.

During the exercise and recovery phases, the NW group showed higher ΔO_2_Hb and ΔHHb values than the OW group. The significances of the group effects were large (0.321 and 0.315, respectively). Cavuoto et al. observed a higher ΔO_2_Hb and ΔHHb in post-repetitive incremental lifting exercises in non-obese participants than in obese participants [32]. Hallmark et al. observed that blood vessel dilation was maintained in normal-weight subjects until 4 weeks after exercise, whereas the obese group showed non-significant vessel dilation through the post-exercise phase [33]. Mechanisms that influence cerebral oxygenation include arterial blood gases, central hemodynamics, cerebral metabolism, and neural mechanisms, including extrinsic autonomic and sensory nerves and intrinsic neurons, which are closely associated with the vasculature within the brain parenchyma [7]. Cerebral oxygen delivery is dependent on cerebral perfusion pressure (CPP) and is inversely proportional to cerebrovascular resistance (CVR) [34]. The fatty tissue releases adipocytokines which induce insulin resistance, endothelial dysfunction, hypercoagulability, and systemic inflammation in obese individual [34,35]. These pathological processes in obesity result in increased arterial stiffness, impaired cerebral endothelial function, and reduced cerebral blood flow and oxygenation [35]. We also observed higher SVRi, an indicator of systemic vascular resistance, in the OW group during the post-exercise period when compared to the NW group. High vascular resistance decreases cerebral blood flow, perfusion, and, consequently, O_2_Hb [35]. A decreased cerebral oxygenation status may increase the risk of cognitive dysfunction at a later age [35,36]. Our study showed that OW subjects had a reduced ability to maintain cerebral oxygenation in the post-exercise phase. This should be considered when prescribing exercise programs to overweight populations. 

This study has several limitations. First, this study had a relatively small sample size, which may have limited our ability to interpret the results. However, the trends in both cardio- and cerebral hemodynamics were consistent with those reported in previous studies. Second, cerebral oxygenation was assessed noninvasively using NIRS at the left prefrontal area level, which implies a very limited spatial resolution and relatively superficial brain tissue measurement. Therefore, our measurements may differ from other gold-standard measurements of cerebral oxygenation (e.g., transcranial Doppler ultrasonography) or measurements from other brain regions. In addition, hematological factors and inter-individual variability in NIRS could also influence our results during the exercise and recovery phases. Future studies with gold-standard measurements of cerebral oxygenation and larger sample sizes are required in order to validate the role of overweight and obesity on cerebral hemodynamic response during and after exercise. Third, the grouping of subjects was based on their BMI status, as suggested by the WHO. However, BMI cannot distinguish between body fat and lean body mass. Studies have reported that waist circumference (WC), waist–hip ratio (WHR), and body fat percentage (BF) measures should be assessed along with BMI, because increasing evidence supports visceral adiposity and/or central obesity as markers of cardiovascular risk [4]. In Europe, a WC of >94 cm and a WHR of >0.9 in men is defined as central obesity [37]. In our study, the mean WC was 100.3 cm and WHR was 0.92 in OW, which indicates that OW subjects may fit the criteria for central obesity.

## 5. Conclusions

In our study, overweight individuals had lower cardiorespiratory fitness than non-overweight individuals. In addition, overweight participants demonstrated lower cerebral oxygenation in the post-exercise phase than non-overweight subjects. A reduced cerebral oxygenation status may be related to limited cardiovascular function in the overweight population. Our study provides information on the cardiovascular and cerebral hemodynamic status during and post-exercise. Thus, clinicians and healthcare professionals should be cautious when prescribing exercise for overweight populations.

## Figures and Tables

**Figure 1 healthcare-11-00923-f001:**
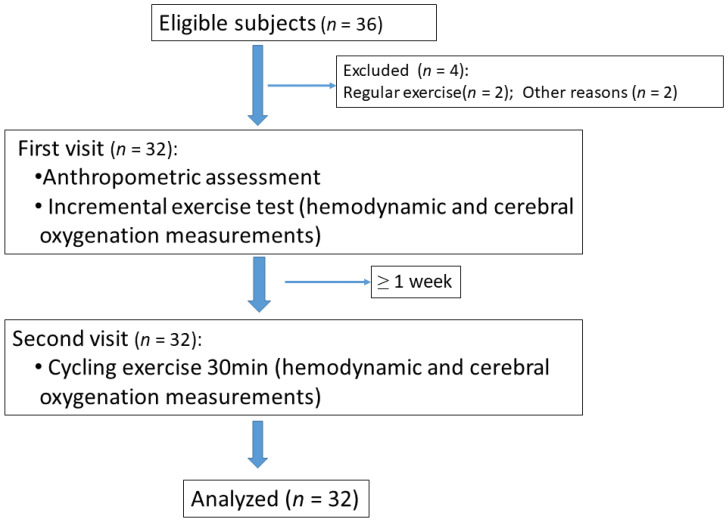
STROBE flow diagram.

**Figure 2 healthcare-11-00923-f002:**
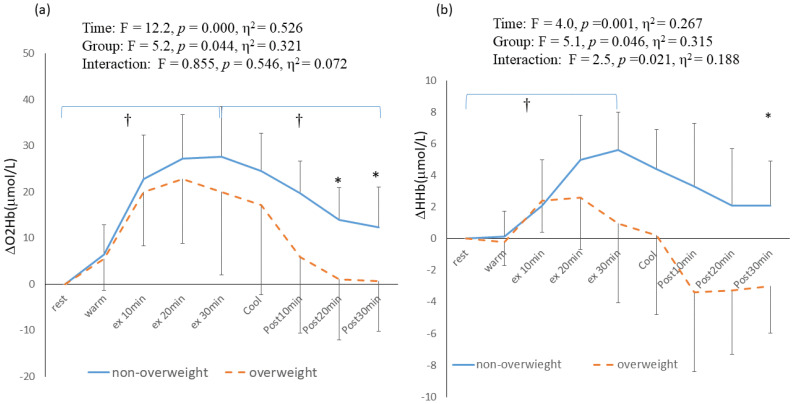
Cerebral oxygenation responses during the 30 min continuous exercise and post-exercise period; (**a**) cerebral oxygenated hemoglobin (Δ O_2_Hb); (**b**) cerebral deoxygenated hemoglobin (ΔHHb). Ex: exercise. †: significant difference within group (*p* < 0.05). *: significant difference between group (*p* < 0.05).

**Figure 3 healthcare-11-00923-f003:**
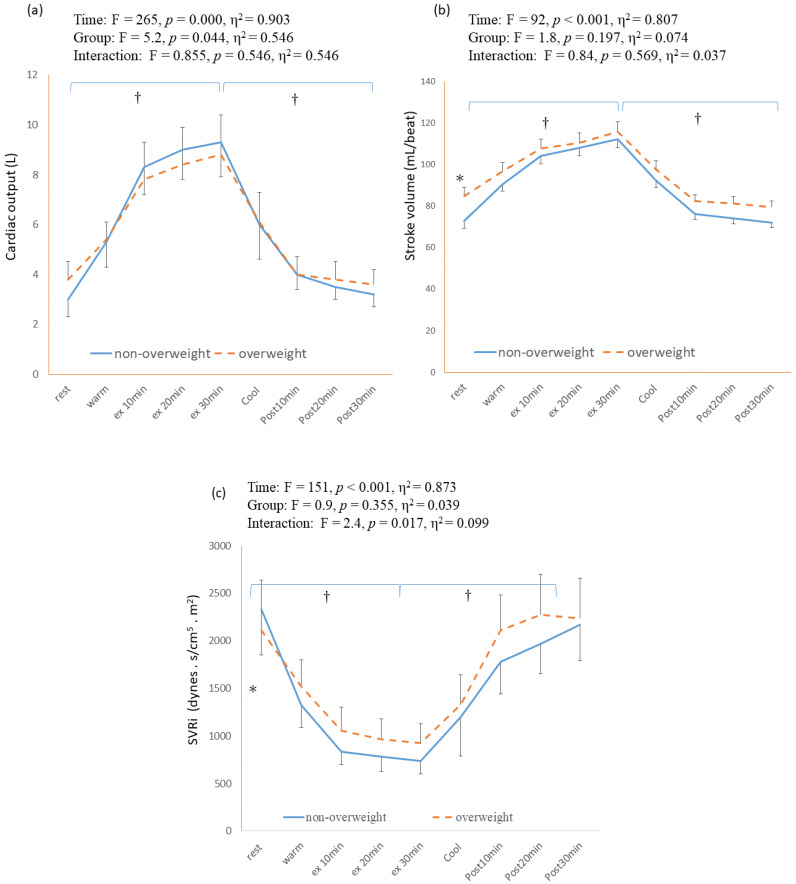
Hemodynamics responses during 30 min of continuous exercise and the post-exercise period. (**a**) Cardiac output; (**b**) stroke volume; (**c**) systemic vascular resistance index (SVRi). Ex: exercise. * *p* < 0.05, between-group difference. †: significant difference within group (*p* < 0.05).

**Table 1 healthcare-11-00923-t001:** Characteristics of participants.

Variables	NW Group (n = 18)	OW Group (n = 14)	*p*
Age (year)	21.6 ± 0.7	21.8 ± 1.0	0.543
Gender (Males) (n)	18	14	
Body height (cm)	174.8 ± 5.2	169.3 ± 6.8	0.021 *
Body weight (kg)	68.5 ± 4.0	83.3 ± 17.5	0.010 *
BMI (kg/m^2^)	22.4 ± 1.4	29.0 ± 5.4	0.001 **
Waist circumference (cm)	82.7 ± 3.3	100.3 ± 13.8	0.002 **
Hip circumference (cm)	99.6 ± 4.5	108.3 ± 13.8	0.006 **
Waist/hip ratio	0.84 ± 0.03	0.92 ± 0.7	0.001 **
Muscle mass (kg)	31.1 ± 1.9	32.0 ± 4.4	0.475
Fat mass (kg)	13.4 ± 3.8	26.5 ± 12.5	0.003 **
Body fat (%)	19.3 ± 4.8	30.6 ± 7.8	0.000 **

NW: normal weight group; OW: overweight group; BMI: body mass index. * *p* < 0.05; ** *p* < 0.01.

**Table 2 healthcare-11-00923-t002:** Incremental exercise test data of participants.

	NW Group	OW Group	*p*
Rest			
HR (bpm)	76.0 ± 12.1	82.6 ± 9.8	0.174
SBP(mmHg)	121.2 ± 9.7	128.7 ± 12.4	0.130
DBP (mmHg)	76.2 ± 8.3	88.4 ± 9.1	0.004 **
At anaerobic threshold (AT)			
Powers (watt)	87.2 ± 26.8	64.7 ± 20.1	0.032 *
VO_2_ (mL/min)	1274.5 ± 283.4	1043.1 ± 254.6	0.049 *
VO_2_/predicted (%)	47.6 ± 6.1	47.3 ± 11.4	0.933
At peak effort			
HR (bpm)	178.6 ± 5.2	167.9 ± 7.0	0.000 **
RPE	18.5 ± 0.51	18.5 ± 0.51	0.864
Powers (watt)	186.0 ± 26.6	153.4 ± 19.6	0.003 **
VO_2_peak (mL/min)	2773.5 ± 410.4	2451.8 ± 283.6	0.019 *
VO_2_peak (mL/min/kg)	41.3 ± 5.7	30.0 ± 5.0	0.000 **
%VO_2_peak predicted (%)	80.1 ± 14.3	66.0 ± 7.5	0.008 **

HR: heart rate; SBP: systolic blood pressure; DBP: diastolic blood pressure; VO_2_: oxygen uptake; RPE: rate of perceived exertion. * *p* < 0.05; ** *p* < 0.01.

**Table 3 healthcare-11-00923-t003:** Correlation analysis between the exercise test and cerebral oxygenation variables.

		Power (Peak)	VO_2_peak	VO_2peak_/kg
TSI peak	r	0.095	0.041	0.052
	*p*	0.659	0.851	0.808
O_2_Hb peak	r	0.353	0.390 *	0.295
	*p*	0.071	0.044	0.135
HHb peak	r	0.508 **	0.469 *	0.468 *
	*p*	0.007	0.014	0.014
tHb peak	r	0.429 *	0.442 *	0.374
	*p*	0.026	0.021	0.055

TSI: tissue saturation index; O_2_Hb: oxygenated hemoglobin; HHb: deoxygenated hemoglobin; tHb: total hemoglobin; VO_2_peak: peak oxygen uptake. * *p* < 0.05; ** *p* < 0.01.

## Data Availability

Data are available from the first author upon reasonable request.

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
