# Peer review of "Comparison of Hemodynamic and Cerebral Oxygenation Responses during Exercise between Normal-Weight and Overweight Men"

_healthcare, 2023, doi:10.3390/healthcare11060923_

Round 1

Reviewer 1 Report

Generally, this article is interesting. However, some issues still need to be explained by the authors.

On lines 76 - 78, the authors mentioned participants' criteria from BMI measurement. BMI is an easy tool to decide the nutritional status of a person, but it is also tricky because BMI cannot tell us a person's body fat. So, why authors classified the participants based on BMI need to be stated in the Participants section and in the limitations.

in Table 1, the muscle mass of both groups was not significantly different, while the fitness results were significant, you may add some explanation about this in the discussion.

The authors need to add more references that support the discussion part, otherwise, the discussion part will sound like just the authors' assumptions.

Author Response

Thank you for the valuable comments.  We appreciate it very much. For the responses to comments, please check the attached file.

Reviewer 2 Report

* In the abstract, write down the whole word for the abbreviations that appear for the first time (e.g. O2Hb, HHb, SVRi).

* The authors need to be very clear about what the novelty and rationale for the study is and what it will add to the field of sports medicine.

* Lines 74: Study type and design should be reviewed. If the protocol included two groups of participants (who had not engaged in regular exercise within the past 6 months, as stated in the inclusion criteria) and were then exposed to the exercise program, this would be an interventional study, not an observational study. 

* Lines 77: The authors reported that the BMI for overweight/obese group was ≥24 kg/m2. However, for adults, WHO defines overweight and obesity as follows: overweight is a BMI ≥25; and. obesity is a BMI ≥30. Authors must provide a valid reference for this classification. Therefore, avoid using the term "obesity" because none of the data presented indicated obesity of the participants. This issue should be corrected in the title, abstract and throughout the manuscript.

* Lines 90: "All participants completed two visits at least 1 week apart......." This sentence is confusing, Please review & clarify.

* Lines 96: "refrain from strenuous exercise at least 24 h prior each visit" This sentence is confusing. The authors stated in the inclusion criteria that participants had not engaged in regular exercise within the past 6 months. Please explain this discrepancy.

* The manuscript needs a thorough revision. Some phrases in the text of the paper need improvement due to various reasons (style, grammar or the lack of clarity). For example

   Lines 120-124: This sentence is very long and confusing, Please clarify.

   Line 321: "Cerebrovascular blow flow" change blow into – blood.

   Line 335: "observed at rest with and elevated hemodynamic" change and into – an. 

* In all tables: write a footnote explaining all of the abbreviations in the table.

* Results: Rather than simply justifying the results, the results must be interpreted and explained to adequately clarify the conclusions. For examples:

   Line 193: Why was there a significant difference in the participants' height? This may add to the exaggerated difference in BMI between the two groups.

   In the overweight group, the increased cardiac output did not correspond to the increased systemic vascular resistance. The authors must provide a reasonable explanation for this.

* Discussion: needs a comprehensive review, as well as scientific explanations with valid recent references. Where possible, please discuss potential mechanisms behind your observations. You should also expand the links with prior publications in the area.

For examples:

1.     Why stroke volume & cardiac output (resting, during exercise & recovery) were higher in obese individuals?

2.     Line 292: "resulting in an increase in blood viscosity and cardiac preload, which accumulates......" Blood viscosity increases the after-load and not pre-load. The author should check the reference.

3.     Line 293: "cause the left ventricular wall to thicken and limit the systolic function of the heart". First: Increased ventricular thickness impairs relaxation leading to diastolic dysfunction, which will then end in systolic failure. Second: systolic function means cardiac ejection which affect stroke volume & cardiac output.

4.     The authors stated that obese individuals have a thicker left ventricular wall and limited systolic function, so how can this sentence be used to explain the greater stroke volume and cardiac output in overweight participants compared to normal weight?

5.     Line 324: the authors should highlight the physiological factors regulating cerebral blood flow, the impact o f exercise & pathological changes induced by obesity, as this is the main issue of study.

6.     Lines 340-343: Please explain the link between blood volume & obesity, adding more recent studies.

7.     Lines 343-346: This sentence is confusing and contains a contradiction, Please clarify. The authors reported diastolic dysfunction in obese individuals (previously as systolic failure in line 293) and this dysfunction was responsible for the high stroke volume & cardiac output.

* The conclusion should be short and to the point, summarizing the main findings and targeting specific objectives. It needs to be re-written. For example:

   Line 371: "overweight/obese individuals had lower physical fitness than...." what data in your study confirm this?

   Line 374: "overweight/obese participants demonstrated lower cerebral oxygenation vascular resistance in the post-exercise phase". The reported lower vascular resistance is inconsistent with what was found in the results.

   Line 374-376: This sentence is confusing and needs a comprehensive revision.

   I think the conclusion exaggerates the results obtained

Author Response

(The authors gave the same response as above.)

Reviewer 3 Report

The authors propose a study in which they analyse differences in cardiovascular and cerebral oxygenation performance between normal weight and overweight/obese individuals.

Please find below my specific commments.

1.     Line 35 – The effect of obesity on arterial stiffness seems also of importance in this context. Indeed, it has been reported that obesity increases the stiffness of the carotid artery which may have important implication for the blood supply to the brain(Chiesa et al., 2019; Giudici et al., 2020).

2.     Lines 90-91 – This sentence is unclear and syntactically incorrect. Please rephrase.

3.     Line 103 – “Tape measure” should be “measuring tape”.

4.     Lines 109-110 – This part of the method is unclear. Do the authors mean that the power was increased at a rate of 20-30 Watts/min until the participant was exhausted. If so, please rephrase.

5.     Line 112 – I do not understand this statement: “The following data were recorded at a mean of 15 s”. Do the authors mean that this parameters were averaged over recording of 15 s? Please clarify.

6.     Line 118-119 – How was the age predicted maximal heart rate calculated? 220 bpm – age?

7.     Lines 132-133 – This statement is in contradiction with that at lines 94-95. Were patients asked to refrain from coffee and alcohol for 12 or 24h?

8.     Tables 1-3 – Please define all abbreviations at the bottom of the table.

9.     Lines 247-248 and Figure 3c – I am not familiar with the unit d.s.cm-5. Please clarify.

10.  Lines 249-250 – This sentence appears to be missing a verb in the second clause.

11.  Lines 284-285 – This statement may be misleading. SV and CO reached their maximum at 30 min of exercise in an exercise protocol that involved only 30 min of exercise. It is possible that SV and CO would have kept increasing in case a longer exercise regime was prescribed.

12. Lines 138-144 – The methods and formulas to estimate the cardiovascular parameters should be explained in more details.

13. Lines 74-88 and table  – The gender distributions of the two groups. Where these similar? If not, how may this have affected the results?

Author Response

(The authors gave the same response as above.)

Round 2

Reviewer 2 Report

   The manuscript has been greatly improved and almost all comments have been taken into account. 

   Authors should verify that the uploaded manuscript contains all corrections made in the report. (i.e. response to reviewer comments). I found the discussion was the same and the new updated references weren't added even though the author provided the full response. Only the abstract and introduction have been corrected, so be careful.

Author Response

Dear Reviewer: 

Thank you for the comments. We appreciate it very much. For detail information, please check the attached file. 

Thank you.
